# Is the future near or far depending on the verb tense markers used? An experimental investigation into the effects of the grammaticalization of the future

Tiziana Jäggi[1]*, Sayaka Sato[1], Christelle Gillioz[2], Pascal Mark Gygax[1]

1 Department of Psychology, University of Fribourg, Fribourg, Switzerland, 2 Swiss Federal Institute for Vocational Education and Training, Lausanne, Vaud, Switzerland

* tiziana.jaeggi@unifr.ch

**Data Availability Statement:** The data are held on the OSF repository under the following link: https://osf.io/s2axr/.

## Abstract

Psycholinguistic approaches that study the effects of language on mental representations have ignored a potential role of the grammaticalization of the future (i.e., how the future manifests linguistically). We argue that the grammaticalization of the future may be an important aspect, as thinking about the future is omnipresent in our everyday life. The aim of this study was to experimentally manipulate the degree of future time references (i.e., present and future verb tense and temporal adverbials) to address their impact on the perceived location of future events. Across four experiments, two in French and two in German, no effect was found, irrespective of our verb and adverbial manipulations, and contrary to our hypotheses. Bayes factors confirmed that our null effects were not due to a lack of power. We present one of the first empirical accounts investigating the role of the grammaticalization of the future on effects of mental representations. We discuss possible reasons for these null results and illustrate further avenues for future research.

## Introduction

Languages are built with grammatical structures, such as grammatical gender [1] or grammatical aspect [2, 3]. These structures vary across languages [4] and have been shown to affect how we mentally represent these different aspects of our environment [5–8]. Prior research on the effect of future time reference on temporal discounting [9–11] has established that languages that obligatorily mark the future grammatically show stronger future discounting effects, although the design of these studies remains disputed [12, 13]. In the present study, we examine the impact of the grammaticalization of the future, that is, the grammatical manifestations of how to refer to the future, and how these grammaticalizations may impact our representations of future events. This approach differs from other experimental studies within the field of future time reference and temporal discounting [14, 15], in that we try to unravel the possible underlying cognitive processes involved from a psycholinguistic perspective, as delineated in our previous theoretical work [16].

**Funding:** These studies were funded by the Swiss National Science Foundation, Grant # 100014_175955. The funders had no role in study design, data collection and analysis, decision to publish, or preparation of the manuscript.

**Competing interests:** The authors have declared that no competing interests exist.

In a nutshell, we argued that thinking about the future is an activity we engage in, on average, every 16 minutes [17] and that it can affect different mental health outcomes [18]. We therefore regard it as important to examine the link between grammatical manifestations of the future and mental representations of future events. Not only is the future a vital part of our everyday life, but time in general has been shown to be an important dimension for generating and processing mental representations of text and discourse [19]. As mental representations have a perceptual basis [20–22] and grammatical structures can make some aspects of our visual perception more salient [23], we hypothesize that the grammaticalization of the future should influence how we construct grounded representations of future events.

## How is the future linguistically realized?

Talking about the future is inherently different than talking about the past or present, in that there is uncertainty about whether an event will take place or not [24, 25]. In European languages there are different nuances of certainty about future events, these nuances are defined by how likely it is that an event will happen. For example, a scheduled meeting or an intended visit are more likely to happen than a predicted outcome of a horse race [24]. This difference in likelihood is reflected in which linguistic devices are used to mark the future: most European languages use the present tense when referring to a very likely event, but when met with uncertainty in a prediction-like context, languages vary with regards to the linguistic devices used [24, 25]. The general notion of referring to a future event is called Future Time Reference (FTR) and can use a variety of linguistic devices [24, 25]. In this study we mainly focus on temporal adverbials (e.g., words that describe the time frame of an action in a clause such as *tomorrow* [26]) combined with a present or a future tense (e.g., a grammatical construction indicating a temporal context in the future such as 'I *will* go.').

In European languages, FTR varies, to some extent, in the use of linguistic devices in prediction-like contexts, that is, whether a language uses lexical or grammatical structures [24]. In German, for example, the future tense is indicated using a modal verb construction, as in (1).

(1) German

*Morgen <u>wird</u> es regnen.*

Tomorrow will it rain:INF.

'Tomorrow, it will rain.'

In French, the future tense is constructed using an inflectional structure, which means that the future tense is marked within the verb form, as in (2).

(2) French

*Demain, il pleuvra.*

Tomorrow it rain:FUT.

'Tomorrow, it will rain.'

An important difference between the German and French future tense is its obligatory use to signal FTR. German future tense use is not obligatory; this means that German speakers can also use the present tense to talk about the future, and they even do this quite often [27]. In contrast, for French speakers it is more common to use a more obligatory form–at least when written [28]. For this paper, we use the terms high and low degrees of FTR to refer to two different situations, as in Jäggi et al. [16]. When comparing French and German, it refers to whether a language requires an obligatory marking (i.e., high degree of FTR) or not (i.e., low

degree of FTR). More importantly for this paper, within each language, it also refers to whether we use the future tense (i.e., high degree of FTR) or the present tense to talk about the future (i.e., low degree of FTR).

## The effects of different FTR

The difference in degrees of FTR between languages was first used by Chen [9] in his paper on the link between FTR strength and intertemporal choices (i.e., saving money, having retirement assets, or adopting healthy behaviors), where he proposed the Linguistic Savings Hypothesis (LSH). This hypothesis states that speakers of languages with a high degree of FTR (such as French) are less likely to engage in future-oriented behaviors compared to speakers of languages with a low degree of FTR (such as German). The mechanisms suggested for this hypothesis were that speakers of languages with a high degree of FTR perceive the distance to future events as greater and generally less concrete compared to speakers of languages of a low degree of FTR. This is because referring to the future in the present tense leads speakers to perceive the future as if it were occurring in the present (or at least closer to the present). Chen [9] chose to establish a dichotomous criteria between high vs. low degree of FTR languages adapting Dahl's [24] observations on obligatory FTR use in prediction-like contexts, so that a low degree of FTR was attributed to languages that do not require the verb tense to mark the future (e.g., German) and a high degree of FTR attributed to languages that require the verb tense to mark the future (e.g., English). Chen [9] found a significant effect of the degree of FTR on behavioral outcomes (correlations), even when other economic and demographic parameters were accounted for (e.g., socio-economic status or origin of the legal system in the corresponding country).

Other studies followed Chen's [9] approach and showed evidence for the LSH with diverse behavioral outcomes, such as corporate savings behavior [10], corporate responsibility [11], research and development investment [11], environmental behavior and policies [29], pro-environmental attitudes [30], future-oriented policies in general [31], as well as religiosity [32]. Nonetheless, some researchers have raised concerns about the methodological approach of these correlational studies [12, 13]. In particular, by reanalyzing Chen's [9] data but taking into account cultural traits (such as geographical and historical relatedness of languages), the correlations no longer yielded statistical significance [12].

Roberts et al. [12] suggested that experimental designs were better suited to investigate the effect of FTR strength. In fact, researchers that followed an experimental approach did find a difference between German-speaking (low degree of FTR) and Italian-speaking (high degree of FTR) school children in their intertemporal choice preference, in support of the LSH [33]. Chen et al. [15] also compared different degrees of FTR within a language using a time preference task with Mandarin speakers. In their study, however, the findings obtained were not in favor of the LSH [15]. Interestingly, the results showed a trend in the opposite direction, suggesting that participants were more patient about receiving a certain reward when it was presented in the future tense.

The impact of the LSH is still unsettled given the different results reported by various experimental studies [14, 15, 33]. Thus, our goal in the present study is to reassess the LSH and its assumed effects on temporal discounting and delayed gratification from an experimental psycholinguistic approach, as extensively discussed in Jäggi et al. [16]. In using such an approach, not only can we address the issue of causality–and overcome issues related to correlational studies [9–11]–, but we can also examine the underlying cognitive mechanism at the heart of the assumed effect. We predict that the mechanism in question is driven by the fact that the

grammatical structures that refer to the future draw attention to temporal information, particularly that of the future.

The thinking-for-speaking hypothesis coined by Slobin [23] provides a potential foundation of the mechanism by which FTR can exert an effect on the mental representations of the future. The thinking-for-speaking hypothesis states that when we prepare our thoughts to be expressed in language, we need to tailor these thoughts into the grammatical structures that our language provides. In turn, this motivates speakers (and listeners) to attend to particular information, which makes certain concepts conveyed through grammar more or less salient. For example, if someone is thinking about a specific event that will happen three weeks from today and expresses it, they would most likely use some sort of FTR to express their thoughts. Depending on whether, in their language, the future tense is or is not marked within the verb form, the attention of speakers (or listeners) might be differently drawn towards the future. The thinking-for-speaking hypothesis has been tested on other domains such as spatial relations [34], grammatical gender [6, 35] and grammatical aspect [7, 8]. We argue that the highlighted differences between the present and the future as suggested by the thinking-for-speaking hypothesis can influence mental representations of the future.

## Mental representations of the future

We define mental representations of temporal events as grounded in space. This definition is derived from research on grounded cognition [36–38], which states that cognitive processes (such as mental simulations of events) are not computed amodally, but are rather influenced or grounded in the body, in our perceptual system, or in the physical and/or social environment [37]. Mental representations play an important role when we process language, or more specifically discourse [39, 40]. Mental representations of discourse events are constructed during language processing, and are constantly updated when confronted with new information [19]. Apart from incorporating semantic information, mental representations carry perceptual information, which can be accessed when processing language [20–22]. This pertains to concrete concepts such as performed actions [41, 42] or forms of objects [43, 44], but can also be observed when processing abstract concepts, such as emotions [45], speed [46, 47], space [48], and importantly, time [49, 50].

As time is considered an abstract concept that cannot be defined by itself [38], its definition is based on metonymy (e.g., the iterative event of a clock ticking defines time) and more importantly here, on temporal metaphorization [38]. This latter aspect is crucial in the context of our study, in that the use of temporal metaphors essentially transforms time into a concrete concept. In temporal metaphors, time is grounded in space and/or motion [51]. For example, time is "a moving object" or "we move through time" spatially. Although the spatial translation of time into metaphors is quite universal, these metaphors do vary across languages: while both English- and Mandarin-speakers use horizontal front/back spatial metaphors, Mandarin-speakers also commonly use vertical up/down temporal metaphors [52]. These differing space-time orientations are linked to writing conventions [53]. As the writing convention for French and German is from left to right, we can safely assume that time is perceptually mapped horizontally from left to right. Consequently, and as we do in the present study, using a horizontal timeline to track the spatial differences of grounding time related to using differing degrees of FTR seems appropriate.

**The present study.** The aim of the present study is to explore whether spatio-temporal representations of time vary as a function of different degrees of FTR, mainly within languages. We look at two languages that have different degrees of FTR between and within them: French, with a higher degree of FTR compared to German, yet with a possibility to use a lower degree

of FTR within the language and German, with a lower degree of FTR compared to French yet with the possibility to use a higher degree of FTR within the language. To manipulate FTR experimentally, we use a combination of temporal adverbials and tenses (present/future), as we detail in the Method sections. Experiments 1 and 2 explore possible FTR effects in French, whereas Experiments 3 and 4 explore possible FTR effects in German. To look for between language effects, a post-hoc comparison between Experiment 2 (French) and Experiment 3 (German) is also conducted.

We hypothesize that within each language, readers perceptually represent sentences with a lower degree of FTR (i.e., present tense and temporal adverbials to indicate the future) as spatially closer to the left–representing the present ($T_0$)–than sentences with a higher degree of FTR (i.e., future tense and temporal adverbials to indicate the future). This hypothesis is drawn from the assumption that a higher degree of FTR habitually emphasizes the difference between the present and the future [23] and thus creates perceptually increased distance as time is grounded in space [36]. As a lower degree of FTR is much more common in German than in French, we expect the effect between a low and high degree of FTR in German to be stronger than in French. In other terms, a feeling of novelty when using the future verb tense in German (quite uncommon) might create an even bigger perceptual distance than in French.

## Experiment 1

### Method

**Participants.** Sixty-nine participants were recruited for this study. We recruited 20 French-speaking students via convenience sampling at the University of Fribourg. An additional 49 French-speaking participants were recruited using Prolific (www.prolific.co) [11.12.2018], a webservice specialized in online research. The inclusion criterion was that participants' first language needed to be French. Participants recruited at the University of Fribourg received experimental credits; the Prolific participants received £6.39 per hour for their participation.

We assessed participants' gender, age, student status and language information such as their first language. Further, we asked participants the first three questions from the Language Experience and Proficiency Questionnaire (LEAP-Q, French version; [54]) to check their level of multilingualism ("Please list all the languages you know in order of dominance."/"Please list all the languages you know in order of acquisition (your native language first)") and the amount of time they are immersed in a French surrounding ("Please list what percentage of the time you are currently and on average exposed to each language. (Your percentage should add up to 100%)"). We used the individual language differences to exclude participants that did not meet the inclusion criterion.

The final sample consisted of 30 female and 39 male participants with a mean age of 30.93 years (SD = 10.54). All participants spoke at least one other language–which is very common in Switzerland [55]–and one third of participants reported speaking languages other than French more than 50% of their time.

### Materials and procedure

**Item construction.** We designed 48 sentences that followed a similar pattern: a person starts an event at a given time (e.g., *In six months, Julie will join an international organization*). The names of the persons were taken from the name registry of the Federal Statistical Office and the 24 most common female and male names from the year 1998 (estimated participant age based on student population) were chosen for our sentences. The events described in the

sentences were common events that people experience in their lives, such as starting a new job, moving to a new apartment or finishing a degree.

As the sentences described events at a given time, six adverbials were defined for events taking place in the present (e.g., *en ce moment* [at this moment], *aujourd'hui* [today]) and 16 adverbials for events in the future (e.g., *dans six mois* [in six months], *en juin* [in June]). Adverbials referring to the future covered a time range between six months up to one year in the future. As temporal adverbials can be placed in different positions within a sentence [56], we decided to construct half of the items with adverbials placed at the beginning of the sentence and the other half with adverbials placed at the end/in the middle of the sentence. This was done to control for a possible position effect.

In addition to the 48 critical sentences, another 48 filler sentences were created with the same structure as the critical sentences. The filler sentences served the purpose of varying the time range and including different adverbials that indicated a time range from tomorrow up to six months.

**Scale construction.** To measure the perceptual effects of our experimental sentences, we chose to create a visual analogue scale based on the findings of the spatial representation of time [57]. The scale represented a timeline from left to right with the poles *tout de suite* [right now] to *beaucoup plus tard* [much later] and was translated as a numerical scale from 0 to 100 when analyzed (i.e., the numerical scale was not visible for participants). The scale was presented at the same time as the sentences and for each sentence, participants were instructed to place the event described in the sentence on the continuous timeline.

**Design and procedure.** Critical sentences were presented in three different FTR strength conditions: a) present tense–present adverbial (PP), b) present tense–future adverbial (PF), c) future tense–future adverbial (FF) (see Table 1).

Participants were presented with all three conditions, as a repeated-measure design (i.e., 16 sentences in each condition). To ensure that all sentences were also presented in the three conditions across the experiment, three balanced lists were created. The full list of items used in all experiments can be downloaded on the Open Science Framework (OSF) [58].

The experiment was programmed with Qualtrics, an online survey program [59], so participants could participate on their own computers at home. The study link was distributed among the French-speaking university students via social media, flyers and word of mouth advertising. As few students enrolled in the experiment, we decided to further recruit Prolific participants [60].

When starting the experiment, participants first read the study information and were then asked for consent to participate in the study. After consenting, we asked participants for their demographic information (age, gender and student status) and language related information (first language and LEAP-Q). Then participants received additional information on how to place events on the timeline by clicking with their cursor on the estimated position. We

**Table 1. Example of critical sentences in three different conditions.**

| Condition | Sentence in French [English translation] |
| --- | --- |
| PP | *Aujourd'hui*, Julie **rejoint** une organisation internationale. [Today, Julie joins an international organisation.] |
| PF | *Dans six mois*, Julie **rejoint** une organisation internationale. [In six months, Julie joins an international organisation.] |
| FF | *Dans six mois*, Julie **rejoindra** une organisation internationale. [In six months, Julie will join an international organisation.] |

*Note*. Adverbials are marked using *italics*; tense is highlighted in **bold**.

emphasized that there were no correct or wrong solutions, but rather, we were interested in their spontaneous opinions. Next, participants were randomly presented with one of the three lists containing 96 sentences to be placed on the timeline–each presented after another in a random order. After placing all 96 events on the timeline, participants were informed that the experiment had ended. On average, participants took 16.5 minutes to complete the study.

**Pre-registration.** Experiment 1 was pre-registered on the OSF by the Centre for Open Science [61]. Experiments 2 to 4 are modified versions of Experiment 1. As planned in the pre-registration, mixed effects models were calculated with *Value* as the dependent variable and *Time condition* as well as *Adverbial position* as fixed effects with a maximal random effects structure justified by the design that will converge. Further, we pre-registered to use Bayes factor to determine whether the data were sensitive enough to either detect a null hypothesis or the alternative one. Not included in the pre-registration were the additional analyses of the temporal adverbials in Experiment 1 and the between language comparison in Experiment 3.

**Ethical consent.** Our request for ethical consent with the number 2018–429 to conduct these studies was granted by the ethics committee from the University of Fribourg.

## Results and discussion

Before the analyses, we removed participants that did not finish the study or that mentioned that their first language was not French. We ended up with a sample of 69 participants.

To check our hypotheses, a series of linear mixed models were computed and compared using the package *afex* [62] in R [63]. All linear mixed models used the *Value* (value of the analogue visual scale) as the dependent variable. The final model contained *Time condition* (PP vs. PF vs. FF) and *Adverb position* (Beginning of the sentence vs. End of the sentence) as fixed effects (*Adverb position* was incorporated to control for a potential effect). *Time condition* was added as a random slope per P*articipants* and a random slope per *Items*, and both *Participants* and *Items* were set as random intercepts. Both predictors employed treatment contrasts. The reference level for the predictor *Time condition* was set to the PF condition, so for the effect of *Time condition*, the intercept is that of the PF condition. For the predictor *Adverb position*, the reference level was set to the Adverb at the beginning of the sentence condition. The Kenward-Roger method was used to calculate the p-values for the terms in the mixed model. Due to lack of convergence, and as justified by the design, the random structure for the model was only composed of *Participants* and *Items* as a random intercept effect. The results of the final model are summarized in Table 2.

The mean and confidence interval of the different conditions are presented in Fig 1. The mean value for the PP condition was M = 2.74, this value was significantly different from the PF (M = 66.67) and FF conditions (M = 66.42). The comparison between the PF and FF condition did not yield significant differences. Random effects showed that the greatest variability derived from residual variability (Var = 181.24, SD = 13.46), which cannot be attributed to either *Participants* (Var = 89.47, SD = 9.46) nor *Items* (Var = 17.67, SD = 4.20).

**Table 2. Summary of the final model:** *Time condition* **and** *Adverb position* **as fixed effects,** *Participants* **and** *Item* **as random intercept effect.**

| Model / Fixed effects | Estimate (β) | df | t-value | p(>|t|) |
|---|---|---|---|---|
| value ~ Time condition + Adverb position + (1\|item number) + (1\|participants) | | | | |
| Intercept (PF) | 66.40 | 123.00 | 47.43 | < 0.001 |
| Time condition (PP) | -63.90 | 3194.37 | -111.42 | < 0.001 |
| Time condition (FF) | -0.27 | 3194.37 | -0.46 | 0.64 |
| Adverb position (end of sentence) | 0.54 | 899.40 | 0.75 | 0.45 |

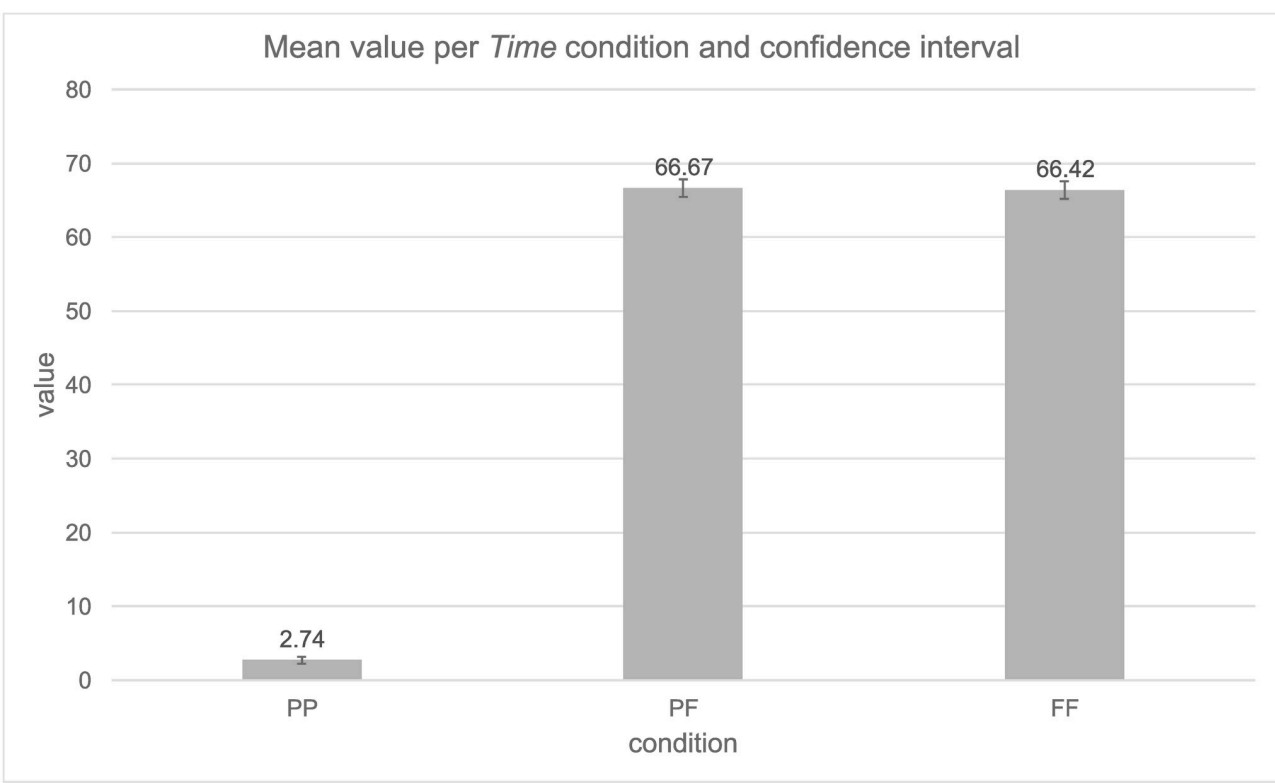

**Fig 1. Mean value per *Time condition* with confidence intervals (95%).**

As the expected difference between the PF and FF did not seem to emerge, we calculated Bayes factor on the lack of effect to assess the relative strength of our evidence. In other terms, we attempted to verify that our data were sufficiently sensitive to detect and strongly support H0 (no effect) over H1 (effect of verb tense) [64, 65]. In order to determine the evidence for H0 over H1 and calculate a Bayes factor, a plausible range of effect is needed. We decided to set it to 32, as it means that PF (expected 34.42) would fall somehow in between PP (2.74) and FF (66.42). We used a half-normal distribution to calculate our Bayes factor, to avoid favoring the probability of supporting H0 over H1 (i.e., half-Cauchy distribution; [66]). In other terms, we statistically tested whether our lack of difference between PF and FF constituted evidence for H0. To do this, we used the difference of -.25 as our sample mean (PF: 66.67; FF: 66.42, and SE = -.46, i.e., the raw difference divided by the *t*-value given by our model encompassing *Time condition*). Using the conventional cut-off of .30 suggested by Jefferys [67] the resulting Bayesian analysis showed strong evidence for the null hypothesis over the existence of an effect of *Time condition B* = .012. This Bayes factor can be taken as substantial evidence for the null hypothesis over the alternative hypothesis (i.e., the alternative hypothesis is .012 times more likely than the null hypothesis). In other terms, our data were sensitive enough to evaluate that the null hypothesis is extremely likely.

To ensure that the adverbials did not influence the placement of events on the timeline an additional analysis was conducted. The analysis of the adverbials revealed an interesting pattern (see Fig 2). Concrete adverbs that mentioned a specific number (e.g., *in six months*, *in eleven months*), followed a discrete distribution (i.e., the bigger the number, the bigger the distance on the scale) suggesting that participants used the numbers in the adverbials to create a mental numerical scale and probably mainly focused on the adverbials rather than the tense.

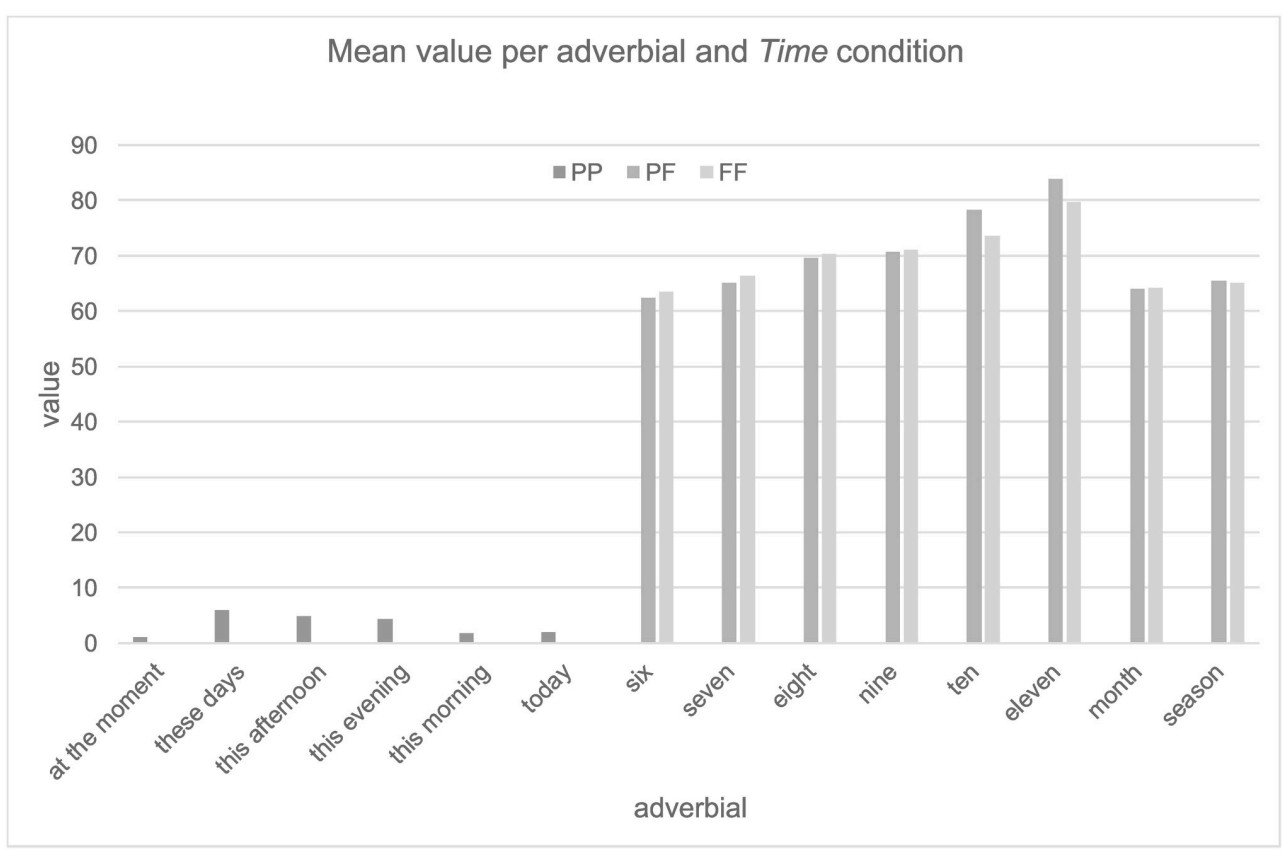

**Fig 2. Mean value per adverbial and *Time condition*.** The numerical adverbials (indicated by the numbers six to eleven) show a distinct pattern reminiscent of the SNARC effect [68]. The pattern for other adverbials is less distinct.

This finding is reminiscent of the SNARC effect (i.e., Spatial Numerical Association of Response Codes), which emerges when participants have to classify random Arabic numbers presented on a screen by clicking either the left or right button on a keyboard. They respond more quickly to small numbers with the left hand and more quickly to large numbers on the right hand, suggesting a spatial mental representation of numerical series [68].

In all, the results from Experiment 1 did not support our within language hypothesis. We found that although participants constructed different temporal representations when reading the present tense to refer to the present (PP) and the future tense (FF), there were no differences in their representations when reading the future tense (FF) and present tense to refer to the future (PF). Interestingly, however, the results of our additional analysis on the adverbials indicated a preference for using concrete, numerical adverbials for participants to decide the position of an event on the timeline. To examine whether this effect was masking a possible effect of tense, in Experiment 2 we modified the temporal adverbials to avoid using concrete ones in order to direct participants' attention to the verb tense.

## Experiment 2

### Method

**Participants.** Sixty-five French-speaking participants were recruited using convenience sampling. Participants were recruited from the student population at the University of Fribourg. Participants' mean age was 24.86 years (SD = 6.42) and the sample consisted of 40

females, 24 males and 1 other gender. All participants reported knowing at least two other languages. Participants from the University of Fribourg received experimental credits for their participation.

## Materials and procedure

**Item construction.** The same 48 items as in Experiment 1 were used in Experiment 2, with the difference that we modified the adverbials to avoid using numerical, concrete adverbials. The new set of adverbials consisted of four adverbials to refer to the present (e.g., *en ce moment* [in this moment], *maintenant* [now]) and eight adverbials to refer to the future (e.g., *d'ici quelque mois* [within a few months], *le semestre prochain* [next semester]).

**Scale construction.** We adjusted the poles of the analogue visual scale from *tout de suite* [right now] and *beaucoup plus tard* [much later] to *proche dans le temps* [close in time] and *éloigné dans le temps* [distant in time]. These changes were considered due to the large dispersion of the data points in the PP condition indicating a semantic ambiguity of the original poles. The new poles emphasized the spatial dimension of time.

**Design and procedure.** No other changes were made, neither to the procedure nor to the experimental design.

## Results and discussion

We explored the within language hypothesis for the French-speaking participants. We used the same parameters for the mixed effects model as in Experiment 1. Again, a model with *Participants* and *Items* as random slopes and intercepts, and *Time condition* and *Adverb position* as fixed effects did not converge. However, the final model did, with *Value* as a dependent variable, *Time condition* and *Adverb position* as fixed effects and P*articipant* and *Item* as random intercepts. The results of the final model are summarized in Table 3.

Mean and confidence interval (95%) are presented in Fig 3. The mean for the PP condition (M = 1.52) was significantly different from the PF (M = 56.29) and FF (M = 56.69) conditions. The two future conditions PF and FF did not significantly differ from each other. Again, for the random effect the greatest variability was derived from residual variability (Var = 180.88, SD = 13.45), which cannot be attributed to either *Participants* (Var = 74.43, SD = 8.63) or *Items* (Var = 62.53, SD = 7.91).

Again, we chose to conduct a Bayesian analysis to check if our data were sensitive enough to support the H0. For this analysis, we calculated the SE = 0.8, the difference between PF and FF diff = 0.4. For this experiment we set the expected value to 28, as the range was smaller compared to Experiment 1. The obtained Bayes factor B = 0.036 indicated that our data were sensitive enough.

The results revealed, again, no significant difference between the PF and the FF condition for our participants. One interpretation was that French may actually not be the ideal language to test our hypothesis, as PF may only be partly present in French and mostly in spoken

**Table 3. Results of the mixed effects model of the French-speaking participants for Experiment 2.**

| Model / Fixed effects | Estimate (β) | df | t-value | p(>|t|) |
|---|---|---|---|---|
| value ~ Time condition + Adverb position + (1|item number) + (1|participants) | | | | |
| Intercept (PF) | 56.16 | 86.40 | 28.14 | < 0.001 |
| Time condition (PP) | -54.84 | 3007.51 | -92.72 | < 0.001 |
| Time condition (FF) | 0.29 | 3007.51 | 0.50 | 0.62 |
| Adverb position (end of sentence) | 0.37 | 46.00 | 0.16 | 0.87 |

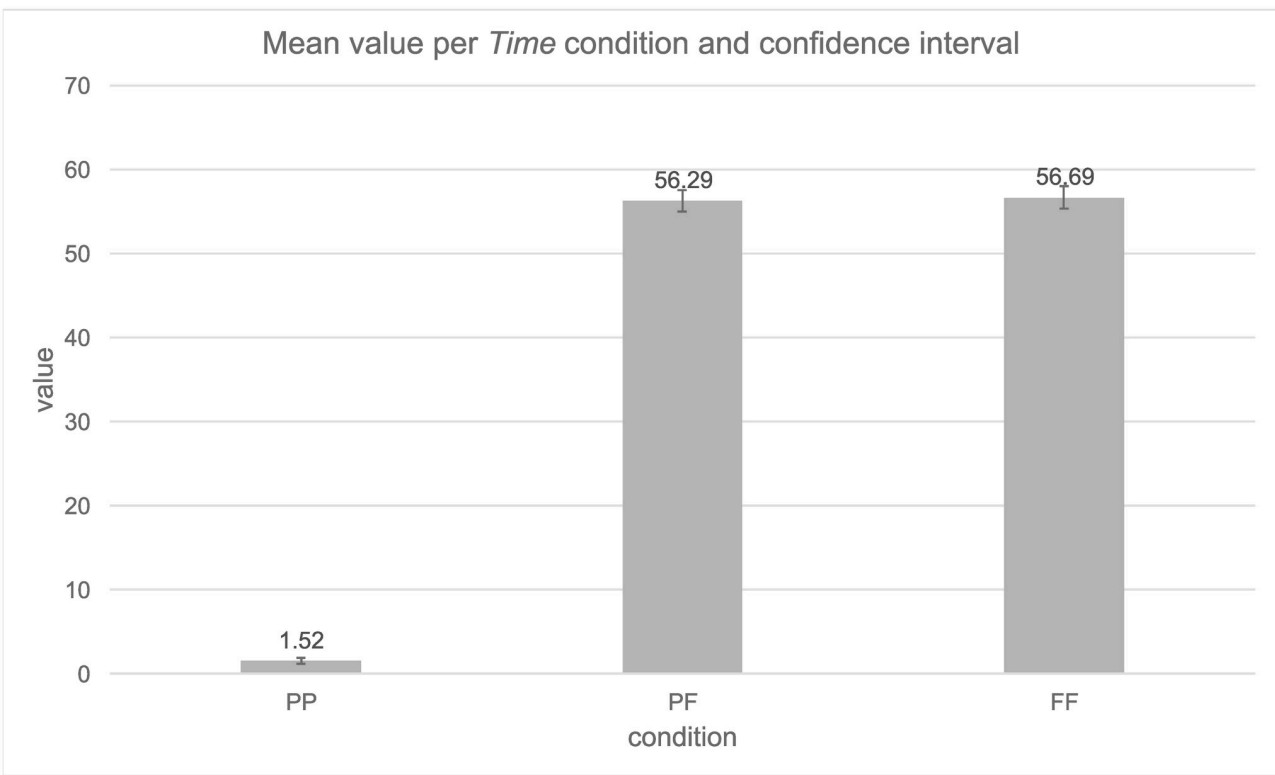

**Fig 3. Mean value per *Time condition* and confidence intervals (95%).**

language. As such, French-speakers may not be as familiar with reading about the future in the present tense, at least not as familiar as German-speakers [16]. A potential novelty effect could have masked a potential FTR effect. In the following experiment, we opted to examine German, as German-speakers are more familiar with speaking *and* reading about the future in the present tense, as well as reading in the future tense.

## Experiment 3

### Method

**Participants.** Sixty-four German-speaking participants were recruited using a convenience sampling method at the University of Fribourg campus. Further, a mailing list from a Swiss German-speaking university was used to advertise the study among German-speaking psychology students. Participants from the University of Fribourg received experimental credits for their participation. We recruited 51 female and 13 male participants with a mean age of 25.25 years (SD = 7.18). All participants reported speaking at least one second language.

### Materials and procedure

**Item construction.** The French items from Experiment 2 were translated by a professional translator into German. The first author translated the items back to French to check for consistency and semantic accuracy. The German equivalent of adverbials to refer to the present were, for example, *im Moment* [in this moment] and *jetzt* [now]) and examples of adverbials to refer to the future were *in einigen Monaten* [within a few months] and *nächstes Semester* [next semester]).

**Table 4. Results of the mixed effects model for the German-speaking participants of Experiment 3.**

| Model / Fixed effects | Estimate (β) | df | t-value | p(>|t|) |
|---|---|---|---|---|
| value ~ Time condition + Adverb position + (1\|item number) + (1\|participants) | | | | |
| Intercept (PF) | 59.74 | 111.93 | 31.61 | < 0.001 |
| Time condition (PP) | -57.41 | 2961.88 | -89.52 | < 0.001 |
| Time condition (FF) | -0.86 | 2961.88 | -1.34 | 0.179 |
| Adverb position (end of sentence) | 0.24 | 46.00 | 0.15 | 0.88 |

**Scale construction.** The scale used in Experiment 2 was translated to German. The German translation of the poles were *zeitlich nahe* [close in time] and *zeitlich entfernt* [distant in time].

**Design and procedure.** No other changes in the design or procedure were made compared to Experiment 2.

## Results and discussion

The same analyses as in Experiment 2 were conducted for the German-speaking participants. Also, the same parameters were chosen for the linear mixed model analysis. Again, the model with the random effect structure with *Participants* and *Items* as random slopes and intercepts did not converge, so the final model used *Participants* and *Items* as random intercepts. The results of the mixed effects model are presented in Table 4; the mean and confidence interval are presented in Fig 4.

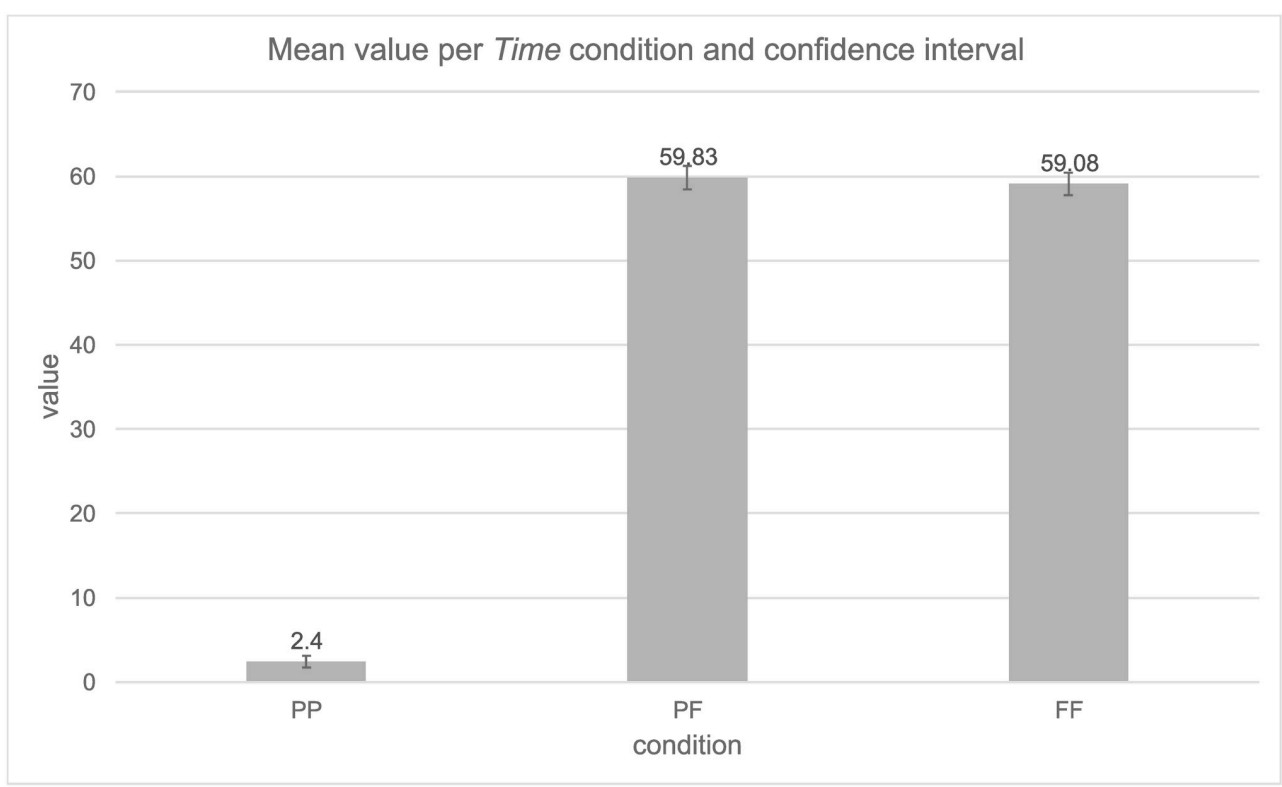

**Fig 4. Mean value per *Time condition* and confidence intervals (95%).**

The PP condition ($M$ = 2.4) was significantly different from the PF and FF conditions. The PF ($M$ = 59.83) and FF ($M$ = 59.08) conditions did not differ significantly. For random effects, the greatest variability was derived from residual variability (Var = 209.71, SD = 14.48), which cannot be attributed to either *Participants* (Var = 137.53, SD = 11.73) or *Items* (Var = 27.58, SD = 5.25).

A Bayesian analysis was conducted with the difference -0.75, SE = 0.56 and the expected effect = 28. The Bayes factor B = 0.006 confirmed our data to be sensitive enough to detect H0.

Additionally, a post-hoc between-language comparison (Experiments 2 and 3) was conducted. Following the grounded theory approach, languages that mark the future obligatorily would make a larger spatial distinction between lower and higher degrees of FTR. For our experimental setup, that would mean that French-speaking participants would generally perceive the future conditions (PF and FF) as spatially more distant than German-speaking ones.

We assessed this hypothesis using a linear mixed model. Again, *Value* was taken as a dependent variable. With regards to the *Time condition*, only the two future conditions (PF and FF) were compared as we did not suspect a difference in the present condition between the two languages. We also took *Language* as a fixed factor, which simply indicated whether the participants were French- or German-speaking. We used the R package *afex* [62] and set treatment contrasts with French as the reference level. We used a random slopes and intercepts structure of *Participants* and *Items*, which converged. *Language* did not seem to significantly explain variance in the data, the analysis can be found on the OSF under supplementary material [58].

As for Experiments 1 and 2, in French, although there was a large difference between PP and FF, as expected, there was no difference between PF and FF. As such, the effect that we expected was not just blurred by a lack of familiarity of PF in French. Now, it could be the case that due to the large difference between the PP condition and the other two conditions, an existing effect between PF and FF could be hidden by creating a temporal zoom-out effect. To examine this final possibility, in the final experiment, we removed the PP condition to zoom in on the expected effect between PF and FF. For this final experiment, we maintained German as our focus language, as we assumed the effect to be bigger in German (see also our rationale for moving from French to German at the end of the discussion section of Experiment 2).

## Experiment 4

### Method

**Participants.** Fifty-four German-speaking participants were recruited for Experiment 4. Participants were recruited using convenience sampling on campus and via social media. As in Experiment 3, the inclusion criterion for participants was defined as German as a first language. Participants consisted of 40 females and 14 males who were, on average, 31.30 years old ($SD$ = 13.92). In total, 52 out of 54 participants spoke at least one second language.

### Materials and procedure

**Item construction.** We kept the same 48 critical PF and FF items as in Experiment 3, removed the PP items, and also eliminated four future adverbials that were tied to a specific month or season. To get a better zoom in effect, we also removed the adverbials in filler items that were too close to the present condition (e.g., *morgen* [tomorrow]).

**Scale construction.** The scale was not changed with regards to Experiment 3.

**Design and procedure.** Again, for this experiment, we dropped the PP condition. So, each participant saw 24 sentences in the PF and FF conditions. Participants were randomly appointed to one of the two lists. No other changes to the design and procedure were made.

**Table 5. Results of the fixed effect for mixed effects model with German-speaking participants for Experiment 4.**

| Model / Fixed effects | Estimate (β) | df | t-value | p(>|t|) |
|---|---|---|---|---|
| value ~ Time condition + Adverb position + (1|item number) + (1|participants) | | | | |
| Intercept (PF) | 53.55 | 96.40 | 16.81 | < 0.001 |
| Time condition (FF) | 0.53 | 2491.42 | 0.99 | 0.32 |
| Adverb position (end of sentence) | 1.40 | 46.00 | 0.42 | 0.68 |

## Results and discussion

To check the within language hypothesis, we conducted a mixed effects model. We used the same parameters as in the experiments before. Initially, the model contained *Value* as the dependent variable, *Time condition* and *Adverb position* as fixed effects and *Participants* as well as *Items* as random slopes and intercepts. As this model did not converge, the final model used *Participants* and *Items* as random intercepts. The results of the mixed effects model can be found in Table 5.

The mean and confidence interval (95%) are depicted in Fig 5. The two future conditions PF (M = 54.21) and FF (M = 54.83) did not significantly differ from each other. The greatest variability from the random effects was derived from *Participants* (Var = 245.6, SD = 15.67), following the residual variability (Var = 185.7, SD = 13.63) and *Items* (Var = 129.5, SD = 11.38).

We conducted a Bayes analysis to evaluate our null result. We used diff = 0.62, SE = 0.62 and the expected value of 28, which resulted in a Bayes factor B = 0.05. So, our data were sensitive enough to confirm H0.

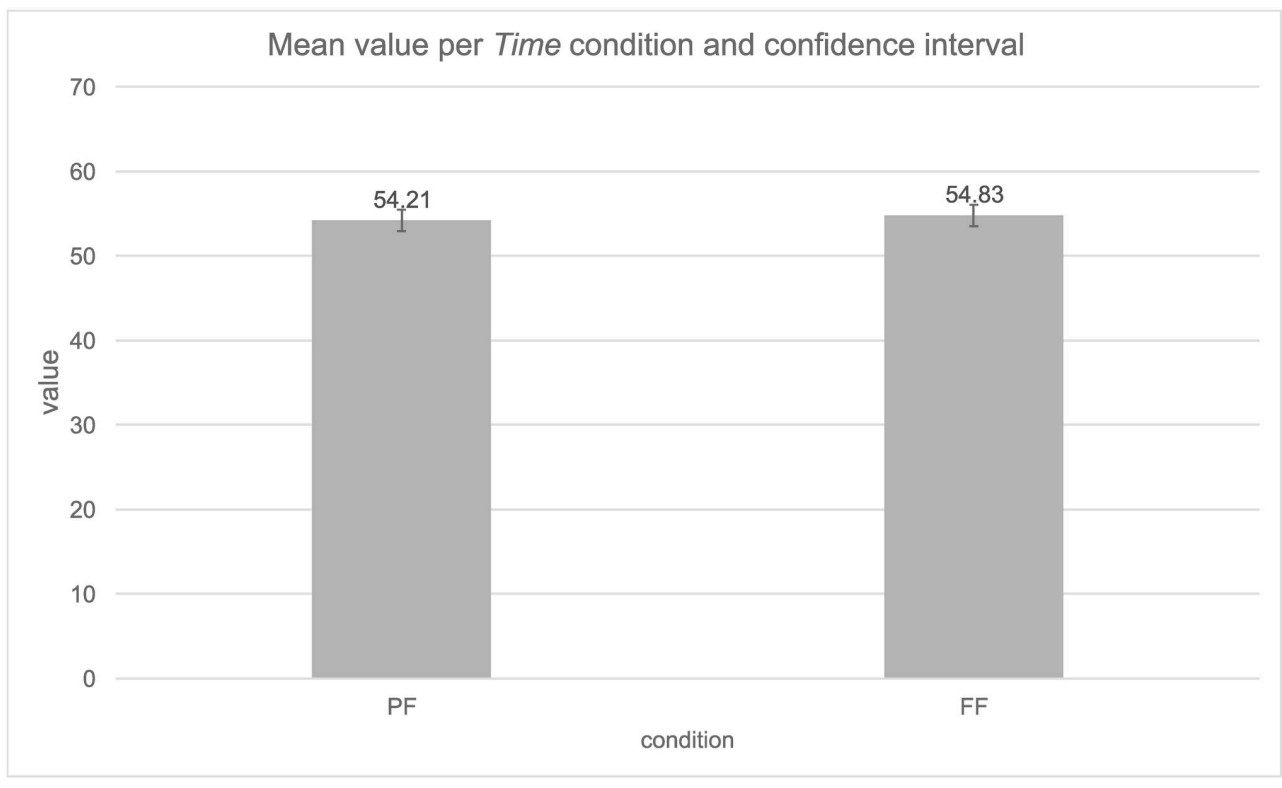

**Fig 5. Mean value per *Time condition* and confidence intervals (95%).**

The results from the mixed effects model show that our conditions were not suitable to explain the variance in the data, which was supported by the lack of significant difference between the two conditions. As such, removing the PP condition to zoom in any effect between the PF and FF conditions did not result in any difference compared to Experiment 1, 2 or 3.

## General discussion

The series of experiments presented in this paper examined the effect of differing degrees of FTR within languages on the grounded representations of future events. We presented four experiments, to test the hypothesis of whether readers ground sentences with lower degrees of FTR as spatially closer to the left–representing the present ($T_0$)–compared to sentences with higher degrees of FTR within a language. The results of the experiments refuted the proposed hypothesis. To confirm that our results were not due to data insensitivity (associated to some lack of power), we also calculated Bayes factors, as advocated by Dienes [64]. These analyses suggested that the null results are more likely true negatives than false ones, especially since all experiments show consistent results. This opens the discussion to analyzing the underlying reasons for these null results.

Could methodological issues account for our findings? As already mentioned, we made consecutive adjustments to the experiments to rule out methodological issues within our proposed methodological setup. Nevertheless, we chose the experimental setup based on assumed spatial differences in mental representations based on grounded cognition induced by varying degrees of FTR. More precisely, as the thinking-for-speaking framework focuses on visual attention as a cognitive mechanism [23], we expected to measure visual-perceptual traces (i.e., perceived spatial location on a timeline). It could, however, be that the shift of attention induced by differing degrees of FTR may have no incidence on mental representations of the future (although it may have an effect on other cognitive mechanisms).

Another methodological issue could have been that the influence of temporal adverbials overpowered the assumed differences in mental representations affected by FTR variations. As mentioned earlier (see the discussion section of Experiment 1), the additional finding on concrete/numbered adverbials is reminiscent of the SNARC effect found in tasks where participants have to classify Arabic numbers presented on a screen by clicking either a left or right button [68]. The SNARC effect is usually found in paradigms measuring response time, although the response pattern suggests that mental representations of numerical or other ordinal sequences, such as days of the week or months of the year, are spatially encoded [69]. This spatially encoded sequence is mainly manifested in working memory [70, 71]. Although, in our task we did not ask participants to memorize ordinal sequences, information may have been encoded in the working memory as new adverbials appeared in subsequent sentences and participants may have implicitly wanted to put new adverbials according to their already answered items. This account could explain why despite reducing the concreteness of the adverbials in the subsequent experiments, participants were still building a mental timeline based on the remaining adverbials. Further research may need to critically evaluate the use of temporal adverbials in similar paradigms to avoid such possible interference.

Should we have focused on cross-linguistic research? In our investigation we examined two languages based on different properties of FTR, but also based on availability as French- and German-speakers are both present in Switzerland, and participants were readily available. Although we included two different languages in our investigation and we conducted a post-hoc between-language analysis, we did not follow the usual cross-linguistic paradigm, where speakers of languages differing more drastically are compared directly. For example,

comparing two languages at the end points of grammatical future marking (e.g., French with obligatory grammatical future marking vs. Finnish with no grammatical future marking) may better document these effects, as those differences may create a more categorically distinct mental representation of the future. Similar research, pertaining to language-and-thought, has examined grammatical and lexical differences that allow clean categorical distinctions [72]. In our experiments we focused on within language differences, which we hypothesized to have perceptual consequences based on fluid grammatical features rather than categorical ones. Perceptual traces of different mental representations may surface with more clean-cut categories. To the best of our knowledge, there has been very little research on such comparisons [16]. Although this may be a legitimate possibility, we would still advocate investigating possible effects with a different paradigm than that tested in the present paper, especially in regards with the lack of a signal of possible perceptual effects.

A recent account, which contends the notion of FTR strength as the responsible driver for temporal discounting, argues that rather modality and, therefore, the amount of certainty transmitted in a FTR may be responsible for probability discounting (rather than temporal discounting) [73]. In order to dismiss the notion that temporal distance is encoded in future tense, Robertson and Roberts [73] conducted a similar experimental paradigm as part of their Study 2. They also experimentally manipulated the present and future tense in English- (high degree of FTR) and Dutch-speakers (low degree of FTR) and used temporal adverbials to invoke different temporal locations on a timeline. Their result matches with the results found in this paper: they could not find a within language effect, where the future tense invoked more temporal distance [73]. The authors argued that their results indicated that FTR strength should not only be determined by future tense, but also by modal variations, which often encode FTR. This is especially true given that their results indicated that the English- and Dutch-speakers in their study were most likely probability discounting, and not temporal discounting. The distinction between future tense and modal variations as attentional cues is important, however, in terms of the thinking-for-speaking hypothesis it also raises two important issues. The first one is related to defining the actual linguistic elements that are most prevalent in FTR, and the second is related to defining what exactly can be considered as future verbal tense. Future research may need to address these intertwined issues to get a more exhaustive picture of the mechanisms at stake.

It may be the case that our hypothesis proved to be wrong, in that different FTR may not have any effect on the way we represent the future. Although our data would suggest this to be true, more data would definitely need to be collected, using different signal traces, maybe linked to other cognitive mechanisms, such as memory encoding (as discussed by Jäggi et al. [16]).

As a concluding comment, we would therefore suggest that future studies focus on both perceptual and non-perceptual effects of different FTR markings, across language groups that have clearer FTR differences. Only then could we have a better perspective on the present null results.

## Author Contributions

**Conceptualization:** Tiziana Jäggi, Sayaka Sato, Christelle Gillioz, Pascal Mark Gygax.

**Data curation:** Tiziana Jäggi.

**Funding acquisition:** Christelle Gillioz, Pascal Mark Gygax.

**Methodology:** Tiziana Jäggi.

**Project administration:** Tiziana Jäggi, Pascal Mark Gygax.

**Supervision:** Sayaka Sato, Pascal Mark Gygax.

**Visualization:** Tiziana Jäggi.

**Writing – original draft:** Tiziana Jäggi.

**Writing – review & editing:** Sayaka Sato, Christelle Gillioz, Pascal Mark Gygax.

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
