## [Decision Letter · Decision Letter 0]

30 Jul 2021

PONE-D-21-18131

Is the Future near or far depending on Verb Tense Markers used? An experimental Investigation into the perceptual Effects of the Grammaticalization of the Future

PLOS ONE

Dear Dr. Jaeggi,

Thank you for submitting your manuscript to PLOS ONE. After careful consideration, we feel that it has merit but does not fully meet PLOS ONE’s publication criteria as it currently stands. Therefore, we invite you to submit a revised version of the manuscript that addresses the points raised during the review process.

Your manuscript has benefitted from 4 reviewers, all experts in this field of research. As you will see, they have provided critical and constructive comments that I think will greatly improve your manuscript, should you choose to resubmit and follow their advice. I have also read your manuscript and find it technically sound, which is a prerequisite for publication in PLOS ONE. There is much to like about the approach you take in your research and generally the results will be useful to other researchers working in this area. However, there are several serious issues pertaining to theory, key terms and concepts, and strength/coherence of argument, identified by the reviewers that would need to be addressed before the manuscript can be considered suitable for publication. Specifically, in a revision, you would engage seriously with the theoretical treatment of the relevant literature (reviewers 2 and 4), disambiguate the use of critical terminology, which currently reflects vague understanding of key terms (e.g. use of the terms ‘perception/perceptual, ‘grammaticized/grammaticization’ etc etc, see comments by reviewers 1, 2, and 3), and offer clear articulation of key concepts and theories to demonstrate adequate understanding and present a more coherent argument (reviewer 2). There are also a couple of serious omissions of previous literature (reviewer 4). I will return the revised manuscript to the reviewers for another round of reviews, so please be sure to address each comment in detail as per the instructions below.

We look forward to receiving your revised manuscript.

Kind regards,

Panos Athanasopoulos, Ph.D

Academic Editor

PLOS ONE

Journal Requirements:

Reviewers' comments:

Reviewer's Responses to Questions

**Comments to the Author**

1. Is the manuscript technically sound, and do the data support the conclusions?

Reviewer #1: Yes

Reviewer #2: Partly

Reviewer #3: Yes

Reviewer #4: Yes

2. Has the statistical analysis been performed appropriately and rigorously? 

Reviewer #1: I Don't Know

Reviewer #2: I Don't Know

Reviewer #3: Yes

Reviewer #4: Yes

3. Have the authors made all data underlying the findings in their manuscript fully available?

Reviewer #1: Yes

Reviewer #2: Yes

Reviewer #3: Yes

Reviewer #4: Yes

4. Is the manuscript presented in an intelligible fashion and written in standard English?

Reviewer #1: Yes

Reviewer #2: No

Reviewer #3: Yes

Reviewer #4: Yes

5. Review Comments to the Author

Reviewer #1: Review Jaggi et al

The authors report that in Experiment 1 participants essentially discriminated between sentences about the present and sentences about the future, irrespective of how the future was written. However, they found a relationship between higher numbers (in the adverbials) and positioning towards the more distant future on the timeline, suggestive of the SNARC effect. This (former) result was repeated across three futher experiments, which modified the type of sentence, whether there were present sentences at all, and which also used German.

I found the hypothesis interesting and sound. I very much liked the methodical step-by-step approach to each consecutive experiment, and each step was entirely merited. Probably like the authors, I was slightly surprised to find no effect of the way the future was grammaticalised on participants' perceptions of temporal distance.

I would suggest expanding a bit more on the reasons why no effect was found. One thing that struck me about the grammaticalisation argument generally is that the lines between 'grammatical', 'modal', 'adverbial', and perhaps even 'lexical' futures are quite blurred, more so perhaps than those traditional areas of language-and-thought research such as colour, grammatical gender, etc (particularly gender). English is a case in point. 'Will' is a modal verb, and the 'going to' future is esentially a present (continuous) tense with an added infinitive. There is, in fact, not real future tense in English. Witness:

The train leaves at 2pm tomorrow.

I'm going to see her later.

I'll see you then.

I'll have been living there for 30 years come February.

Etc.

All of these are modal verbs or present tenses, each of which have alternative meanings that are unrelated to the future, just 'tweaked' with an adverbial that can refer to the future. My understanding of French is that yes, it uses suffixes on verbs like most romance languages, but more English-like options exist, and of course German and English are related anyway. So perhaps the issue is that there is not enough of a 'clean' categorical distinction between (these) languages for the hypothesis in question. I'd recommend adding more to the para beginning on line 415. Another thought is that there are some languages which only use adverbials to refer to the future, I believe (Hebrew comes to mind, though I may be wrong there). Some languages also have 'strong' past tenses, such as Italian. I don't think you need more data for this particular paper, just a thought for the future (pun unintended).

Some other points:

The table of results (Exp 1) seems a bit thin and a tad confusing - please report that the intercept refers to PF. I deduced this after looking at the graph. Lmers in R usually put the alphabetically earliest factor level as intercept but this information didn't help me as you've reorganised the levels. Also, what was the result for the random effects (i.e., how much of the data did they account for)? It would also be useful if you could report any deviations from your pre-registration, which is not accessible at the time of this review.

I am unclear as to the procedure used for the specific Bayesian test used in Exp 1, but this might be because of typos in the values: aren't the correct numbers 66.42 and 66.47 (not 64.42 and 64.47)? It's also a bit unusual to interpret a Bayesian test as showing anything to be 'truly' sensitive; better to say it's just extremely likely that the data were sensitive enough, etc etc. The raw results certainly back this null up. I'd also report what the BFs mean in layman's terms (i.e. the data were x times more likely under the null hypothesis).

Other

line 18: This is a contentious statement to make with references, let alone without.

At the time of reviewing the pre-registration details for Exp 1 were not available.

Abstract line 11 : irrespective > irrelevant

line 244 masking > shadowing

line 386-387: clunky

Reviewer #2: REVIEW OF: Is the Future near or far depending on Verb Tense Markers used? An experimental Investigation into the perceptual Effects of the Grammaticalization of the Future

The authors present a series of four experiments which test the hypothesis that Future Time Reference (FTR) grammaticization impacts how temporally distal people construe future outcomes to be. On the whole, it appears the authors have done some interesting research which undermines a widely-cited account of the semantics of future tenses and their pursuant effects on construals of future events. The authors have developed a new task and tested an interesting hypothesis, and this is commendable. I believe that the empirical work this MS presents has merit and will be published eventually.

However, the presentation of the work is critically flawed and this prevents me from recommending it be published in its present form. This is principally because the manuscript does not give adequate theoretical treatment to the relevant literature, and repeatedly misconstrues and misunderstands key concepts which need to be properly understood and explained for the MS to be convincing. For instance, there is persistent conceptual drift between within- and between-language effects; the introduction is framed in terms of cross-linguistic effects, yet the studies involve within-language methods and the theoretical bridge between these domains is underdeveloped.

Another oddity is that grammaticization is implied to manifest differently in different speakers at different times. For instance, by the account in the MS, when a speaker says It rains this is “weakly grammaticized” and when the same speaker says It will rain this is “strongly grammaticized”. This fundamentally misunderstands grammaticization, which are those language-level processes which see lexical linguistic elements evolve into grammatical ones. Diachronic processes of grammaticization can therefore lead to cross-linguistic differences in the extent to which languages oblige the use of grammatical markers in certain speech contexts. However, the MS collapses differences in speaker-level usage with notions of grammaticization in such a way as to demonstrate that core concepts have been misunderstood. Key references are cited (Bybee, Dahl) but the MS does not in its present form demonstrate that these have been understood. Another example is that “thinking for speaking” effects are invoked as a theoretical motivation, yet the methods do not investigate these. Additionally, the locus of the cognitive effects under investigation does not appear well understood. The MS repeatedly refers to perceptual effects, yet the methods do not investigate perception, and rather focus on explicit judgments of linguistic cues.

The MS gives an incoherent account of how the semantics of temporal adverbials work (tomorrow, next week, etc.). Firstly, it is suggested that use of a temporal adverbial indicates reduced grammaticization, even though temporal adverbials and future tense operate somewhat orthogonally. The contrast between English (strongly grammaticized FTR) and German (weakly grammaticized FTR) is relevant. Even though English obliges the future tense for prediction-based FTR and German does not, it is acceptable in either language to use a temporal adverbial in combination with the future tense. Secondly, temporal adverbials in German and French transparently encode notions of temporal distance so it is unsurprising that participants rate distal adverbials (e.g. nine months) as farther than proximal ones (e.g. one month). Yet the MS treats this as though it is a novel/interesting result. Too much is made of this, which only distracts from the interesting null effect of tense framing on ratings of temporal distance, and the MS is muddled as a result.

Large and critical areas of literature are unreferenced. In particular, there is a large literature which has focussed on Chen’s (2013) hypothesis that the obligation to use a future tense for prediction-based FTR should cause speakers to construe future events as more temporally distal, and that this will therefore lead to cross-linguistic effects of FTR grammaticiation on intertemporal decision making. Numerous studies, including ones whose results bear direct relevance to the present MS, have followed up on Chen’s (2013) hypothesis. Yet none of this literature is cited, even though the MS purports to test an idea which can be directly traced to Chen (2013). Reading the MS in its own right, it is difficult to make out why future tense marking should lead to distal temporal construals, which makes it all the more strange that Chen (2013) is left uncited. Chen (2013) provides a closely reasoned and mathematically presented account of this hypothesis...

Additionally, while the results are interesting, not enough is made of them. For instance, linguists continue to debate the semantics of FTR, and the future tense in particular, and many of these debates revolve around the entanglement modal with temporal notions in future tense semantics. Given the results in the MS indicate there is no effect of tense framing on ratings of temporal distance, this literature should be engaged, which it currently is not. In fact, the MS’s treatment of modality is generally underdeveloped, and this should be addressed.

The individual studies are not well motivated. The rationale for each study often does not bear up to close scrutiny, or relies on reference to materials which were not included in the material to be reviewed, i.e. the linguistic task which was developed. This makes judging the substantive contribution and motivation of each study difficult.

This brings up a final point, which is that the methods are not clearly reported. While I know roughly what was done, the precise nature of the empirical work remains a mystery. The methods in their present form would not, for instance, provide enough information for the experiments to be replicated. This obviously needs to be corrected.

I fear this review has been overwhelmingly negative, and I want the authors to know that I do believe they have done some worthwhile work. In this review I attach a copy of their MS which I have annotated with suggestions for how it might be improved, and I hope they will implement these in the future, regardless of the editorial decision from PLOS ONE.

Personally, I am very interested in this work; I am near to completing a PhD which has focussed largely on a very similar research question. Much of my work is presently under review or in prep, but I would be happy to be contacted by the authors should they wish to discuss or share results. I can be

contacted at cole.robertson@ru.nl

Reviewer #3: MS PONE-D-21-18131

Title: Is the Future near or far depending on Verb Tense Markers used? An experimental

Investigation into the perceptual Effects of the Grammaticalization of the Future

Authors: Jaeggi, Gygax, Gillioz, Sato

Summary

This manuscript presents four experiments exploring the role of different types of linguistic futures on perceptual qualities. Experiment 1 tested French speakers on the placement of event descriptions on a provided timeline. The results revealed no differences in the different types of futures explored, contrary to the hypothesis. Experiment 2 was a replication of Experiment 1, except concrete temporal adverbials were not used, and a more categorical response modality was used. The results were like those of Experiment 1. Experiment 3 was another replication, how this one involved the use of German speakers. The results were like those of Experiments 1 and 2. Experiment 4 was a replication of Experiment 3, but without the present tense items. The results replicated the first three experiments in not showing an effect of different types of futures. The results revealed that the perceptual aspect of future thinking is largely unaffected by the type of future language structure used.

Evaluation

This manuscript addresses an interesting topic. The study was well-conduced, and the resulting data are adequately interpreted. Although the study is largely a set of null effects, they do have important theoretical implications, and should be made available. I only have one concern that is detailed below.

Minor point

1. I was not entirely satisfied with the description of “perceptual representations”. I think that a more adequate term for capturing what is being referred to here is “embodied representations.”

Reviewer #4: This is an interesting study that presents some important results. The authors should be commended for preregistering the study, seeking to publish even with null results, and for the application of the Bayesian power estimate. This is good science.

I have some points that may require revision of the manuscript. The two main ones relate to the relationship between time and space, and the relation between this study and some parallel work on future discounting. Neither should prevent publication, but the authors may want to think about revisions to make their claims more clear.

1.

Is it necessary to include the spatial translation of time in the hypothesis? Experiment 1 asks participants to express the distance on a slider ranging from left (low) to right (high), so a link between the grammaticalisation and perception of distance could appear even if the participants thought of time as flowing from right to left. The left/right mapping might also predict that the effect would be weaker if the slider was in the other direction. But this isn't tested. So I wonder if the spatial mapping is a necessary step in the hypothesis?

2.

I found it surprising that the paper did not link to the hypothesis by economist Keith Chen on FTR and perceptions of time:

Chen, M. K. (2013). The effect of language on economic behavior: Evidence from savings rates, health behaviors, and retirement assets. American Economic Review, 103(2), 690-731.

Note that the cross-cultural correlation has been criticised:

Roberts, S. G., Winters, J., & Chen, K. (2015). Future tense and economic decisions: controlling for cultural evolution. PloS one, 10(7), e0132145.

But the original idea provides several specific models for how grammaticalisation in language might affect perception of time. In particular, the idea relies on habitual requirements to make distinctions between the future and the present. That is, the more a specific person makes this distinction in their language, the greater the effect on their thought. (that is, there may be within-language differences).

A summary of this research is presented in this preprint (by Cole Robertson and others):

https://papers.ssrn.com/sol3/papers.cfm?abstract_id=3628501

This also includes a summary of recent experimental work on the link between FTR and time perception. Several of these experiments may be relevant to the current study, and the authors may wish to cite them (especially since they claim that little work has been done).

The preprint above also makes a point about the interpretation of future tense. While many see tense as encoding temporal distance (*when* an event occurs), it also communicates modal possibility (*whether* an event will occur). This is particularly the case for modals (e.g. in English "will" and "might" differ in probability more than they differ in temporal distance).

Cole Robertson, as part of his PhD thesis and a few papers under review, did some studies on FTR and time perception in English, Dutch and German.

One experiment was similar to the present study: Presenting participants with a phrase and asking them to indicate with a slider how far away in time they felt this was referring to. A similar experiment was also done with a slider indicating certainty.

In Robertson's study, within a language, the future tense frame did not predict the rating of temporal distance. This is in line with the current results.

However, the rating of certainty did differ significantly by grammatical form: modals were rated as less likely to occur than the future or present tenses.

Participants in Robertson's study also rated objective times (one month, two months) as a check that participants were rating things sensibly. This also agrees with the analysis of numbers in adverbials in the current study.

In addition, Dutch speakers rated events as more distant than English speakers. A similar thing could be tested in the current study. That is, is it possible to compare the scale scores in the current study for German and French directly? (i.e. the between-language prediction). It seems like the French participants are placing the slider at higher positions on average than the German participants. Isn't this what your hypothesis would predict? I appreciate the items are not exactly the same, but is it still worth noting? I'll also note that the authors are in a good position to do an experiment with bilinguals in several languages, making the comparison more effective (perhaps for future research).

Robertson continued with some experiments to show that an individual person's usage of future vs. modal strategies for talking about probability (collected in a survey similar to Dahl's survey) could be used to predict their attitude to future events (e.g. in a future discounting task).

I appreciate that these studies are not yet publicly available (though I'm sure the author would share their thesis manuscript if asked), and the point is not to undermine the current study. In fact, both lines of research seem to agree in their results. My intention is just to flag this converging evidence, and to ask the current authors whether there is a way of harnessing their current data to investigate the question about modal possibility, in addition to temporal distance.

Minor points:

"These structures vary across languages and have shown to

affect how we mentally represent and perceive different aspects of our environment." - This sentence appears to have no evidence attached. It's not clear whether the citations in the previous sentence cover this. To be safe, there are some seminal works that could be easily cited here.

"The grammaticalization of the future, which represents the grammatical manifestations of how to refer to the future, has been scarcely studied with experimental psycholinguistic methods"

"As perceptual representations, we mostly refer to perceptual representations of

distance" - Does this mean distance in time? Or are you talking more generally about any kind of domain?

Lines 37-38. It's my understanding that "future time reference" refers to the act of talking about the future, rather than the linguistic devices used to do that?

"FTR varies in the degree of grammaticalization across languages (and at times within language): a low degree of grammaticalization is characterized by adverbials and modal verbs; and a higher degree of grammaticalization is characterized by grammatical structures embedded in the verb, like a suffix of the future tense (e.g., in French: ‘j’ irai à Paris’ [I will go to Paris])."

There is a lot of academic work on grammaticalisation from linguistics, and this could be cited here.

The French example is not illustrative unless one understands French. Please give the interlinear gloss in addition to the translation.

"as some languages have been shown to have very low grammaticalized future verb tense and to only use modal verbs to indicate the future tense (e.g., German:"

I'm not sure this is technically correct. German can also use modal modifiers (möglich) or mental state predicates (erwarten). Maybe you're grouping these under modal verbs?

"we expect the effect between low and high degree of FTR in German to be somehow stronger than in French". Why "somehow" - you have an explicit hypothesis?

Table 2 - the significance is easier to interpret if it's explained that PF condition was used as the intercept condition.

Mixed effects models: You are modelling a scale with a floor and ceiling. Does the model take this into account? It looks like there might be floor effects for the PP condition? You could do this with e.g. logit function.

Line 365: "it did not significantly account variance for Value" - some missing words in this sentence? Also, some statistical support for this claim should be added (e.g. difference in variance explained in a model comparison test)

The size of the text in the figures is quite small and may not reproduce well at a smaller scale.

6. PLOS authors have the option to publish the peer review history of their article (what does this mean?). If published, this will include your full peer review and any attached files.

Reviewer #1: No

Reviewer #2: No

Reviewer #3: No

Reviewer #4: No

---

## [Author Response · Author response to Decision Letter 0]

30 Sep 2021

Manuscript ID PONE-D-21-18131

Dear Professor Panos Athanasopoulos,

We are pleased to resubmit our manuscript entitled “Is the future near or far depending on verb tense markers used? An experimental investigation into the effects of the grammaticalization of the future” to PLOS ONE. 

We appreciate the reviewers’ critiques of the initial submission and believe that our manuscript greatly improved by responding to their comments. Mainly, we rewrote parts of the introduction concerning the linguistic background to make our argument clearer and more consistent. Further, we clarified what was referred to as ‘perceptual representations’ by adding some key literature concerning grounded cognition. Finally, we provided the omitted literature concerning the work of Chen (2013) and effects of FTR on temporal discounting. Each comment provided by the reviewers has also been addressed point-by-point in our response letter. Our answers are in italic font to facilitate the distinction between the reviewers’ comments and our response.

We hope that the revisions we have provided have led to a better version of the manuscript and that you find it fitting for publication in PLOS ONE.

Best regards,

Tiziana Jäggi

Response to Reviews: Manuscript ID PONE-D-21-18131

Response to Reviewer 1:

“The authors report that in Experiment 1 participants essentially discriminated between sentences about the present and sentences about the future, irrespective of how the future was written. However, they found a relationship between higher numbers (in the adverbials) and positioning towards the more distant future on the timeline, suggestive of the SNARC effect.”

Thank you for mentioning the SNARC effect. We were not aware of such effect, but learning about it made us realize that it is highly relevant for our paper especially with regard to the results of Experiment 1. This is why in the Discussion of Experiment 1 and the General Discussion we have added some sentences around the SNARC effect to explain our results. Please see lines 418ff and 641ff (the lines refer to the document with tracked changes).

“This (former) result was repeated across three further experiments, which modified the type of sentence, whether there were present sentences at all, and which also used German.

I found the hypothesis interesting and sound. I very much liked the methodical step-by-step approach to each consecutive experiment, and each step was entirely merited. Probably like the authors, I was slightly surprised to find no effect of the way the future was grammaticalised on participants' perceptions of temporal distance.

I would suggest expanding a bit more on the reasons why no effect was found. One thing that struck me about the grammaticalisation argument generally is that the lines between 'grammatical', 'modal', 'adverbial', and perhaps even 'lexical' futures are quite blurred, more so perhaps than those traditional areas of language-and-thought research such as colour, grammatical gender, etc (particularly gender). English is a case in point. 'Will' is a modal verb, and the 'going to' future is esentially a present (continuous) tense with an added infinitive. There is, in fact, not real future tense in English. Witness:

The train leaves at 2pm tomorrow.

I'm going to see her later.

I'll see you then.

I'll have been living there for 30 years come February.

Etc.

All of these are modal verbs or present tenses, each of which have alternative meanings that are unrelated to the future, just 'tweaked' with an adverbial that can refer to the future. My understanding of French is that yes, it uses suffixes on verbs like most romance languages, but more English-like options exist, and of course German and English are related anyway. So perhaps the issue is that there is not enough of a 'clean' categorical distinction between (these) languages for the hypothesis in question. I'd recommend adding more to the para beginning on line 415. Another thought is that there are some languages which only use adverbials to refer to the future, I believe (Hebrew comes to mind, though I may be wrong there). Some languages also have 'strong' past tenses, such as Italian. I don't think you need more data for this particular paper, just a thought for the future (pun unintended).”

Thank you for this remark. Indeed, we mainly looked at grammatical determinants of FTR, rather than lexical or modal ones. We rewrote a part of the Introduction about linguistic features of the future and added on to the General Discussion discussing other possible FTRs to make this distinction clearer. We hope by doing so we have been able to meaningfully incorporate your concerns. Please see lines 68ff and 662ff. 

“Some other points:

The table of results (Exp 1) seems a bit thin and a tad confusing - please report that the intercept refers to PF. I deduced this after looking at the graph. Lmers in R usually put the alphabetically earliest factor level as intercept but this information didn't help me as you've reorganised the levels. Also, what was the result for the random effects (i.e., how much of the data did they account for)?”

Thank you for these comments. As you suggested, we added more information on the models in the respective parts. See lines 368ff. 

“It would also be useful if you could report any deviations from your pre-registration, which is not accessible at the time of this review.”

Thank you for this comment. We made the pre-registration available now. The pre-registration answered questions concerning whether any data had already been collected, what the main question and the hypothesis of the project were, what the key variables were, how many and which conditions participants would be assigned to, which analyses would be conducted and how many observations would be collected. See lines 347ff. 

“I am unclear as to the procedure used for the specific Bayesian test used in Exp 1, but this might be because of typos in the values: aren't the correct numbers 66.42 and 66.47 (not 64.42 and 64.47)? It's also a bit unusual to interpret a Bayesian test as showing anything to be 'truly' sensitive; better to say it's just extremely likely that the data were sensitive enough, etc etc. The raw results certainly back this null up. I'd also report what the BFs mean in layman's terms (i.e. the data were x times more likely under the null hypothesis).”

Thank you very much for this attentive remark. Indeed, these numbers were typos. Further, we removed the word ‘truly’ and addressed your remarks concerning the accessibility of the Bayesian results. See lines 403ff. 

“Other

line 18: This is a contentious statement to make with references, let alone without.

At the time of reviewing the pre-registration details for Exp 1 were not available.

Abstract line 11: irrespective > irrelevant

line 244 masking > shadowing

line 386-387: clunky”

We appreciate these suggestions. You can find the corresponding changes on lines 36f, 347ff, 29, 438 and 621f.

Response to Reviewer 2:

“The authors present a series of four experiments which test the hypothesis that Future Time Reference (FTR) grammaticization impacts how temporally distal people construe future outcomes to be. On the whole, it appears the authors have done some interesting research which undermines a widely-cited account of the semantics of future tenses and their pursuant effects on construals of future events. The authors have developed a new task and tested an interesting hypothesis, and this is commendable. I believe that the empirical work this MS presents has merit and will be published eventually.”

“However, the presentation of the work is critically flawed and this prevents me from recommending it be published in its present form. This is principally because the manuscript does not give adequate theoretical treatment to the relevant literature, and repeatedly misconstrues and misunderstands key concepts which need to be properly understood and explained for the MS to be convincing. For instance, there is persistent conceptual drift between within- and between-language effects; the introduction is framed in terms of cross-linguistic effects, yet the studies involve within-language methods and the theoretical bridge between these domains is underdeveloped.”

Thank you for your important remarks. We rewrote a great part of the introduction, especially, on the linguistic realizations of the future and tried to improve the theoretical bridge between within and between language effects. See lines 68ff and 241ff (the lines refer to the document with tracked changes). 

“Another oddity is that grammaticization is implied to manifest differently in different speakers at different times. For instance, by the account in the MS, when a speaker says It rains this is “weakly grammaticized” and when the same speaker says It will rain this is “strongly grammaticized”. This fundamentally misunderstands grammaticization, which are those language-level processes which see lexical linguistic elements evolve into grammatical ones. Diachronic processes of grammaticization can therefore lead to cross-linguistic differences in the extent to which languages oblige the use of grammatical markers in certain speech contexts. However, the MS collapses differences in speaker-level usage with notions of grammaticization in such a way as to demonstrate that core concepts have been misunderstood. Key references are cited (Bybee, Dahl) but the MS does not in its present form demonstrate that these have been understood. Another example is that “thinking for speaking” effects are invoked as a theoretical motivation, yet the methods do not investigate these. Additionally, the locus of the cognitive effects under investigation does not appear well understood. The MS repeatedly refers to perceptual effects, yet the methods do not investigate perception, and rather focus on explicit judgments of linguistic cues.”

Thank you very much for these comments. Indeed, in the first draft of the manuscript the concept of grammaticalization was not clearly explained and used interchangeably with FTR, creating the theoretical inaccuracy you mentioned. Therefore, we edited a great portion of the introduction addressing your comments and hope to have made this part clearer. Please see lines 68ff.

We still feel that thinking-for-speaking effects are an important part of our theoretical development to understand our experimental setup. However, we do agree that the term ‘perceptual’ creates confusion and a certain misunderstanding regarding the cognitive effects than help make the manuscript clearer. Therefore, we rephrased the concerned parts in the manuscript. Please see lines 182ff.

“The MS gives an incoherent account of how the semantics of temporal adverbials work (tomorrow, next week, etc.). Firstly, it is suggested that use of a temporal adverbial indicates reduced grammaticization, even though temporal adverbials and future tense operate somewhat orthogonally. The contrast between English (strongly grammaticized FTR) and German (weakly grammaticized FTR) is relevant. Even though English obliges the future tense for prediction-based FTR and German does not, it is acceptable in either language to use a temporal adverbial in combination with the future tense. Secondly, temporal adverbials in German and French transparently encode notions of temporal distance so it is unsurprising that participants rate distal adverbials (e.g. nine months) as farther than proximal ones (e.g. one month). Yet the MS treats this as though it is a novel/interesting result. Too much is made of this, which only distracts from the interesting null effect of tense framing on ratings of temporal distance, and the MS is muddled as a result.”

Thank you very much for this remark. We adjusted this part, where it is suggested that use of a temporal adverbial indicates reduced grammaticalization. Again, this inaccuracy was caused by our unconsidered use of the word ‘grammaticalization’. Please see lines 131ff. 

Indeed, it is unsurprising that participants would rate ‘nine months’ as farther than ‘one month’. However, we did not expect to find such a scale-like distribution for concrete numbers. We feel this finding, although less important, needs to be appropriately addressed, especially given that Reviewer 1 had mentioned that it is reminiscent of the SNARC effect. We have therefore developed on this issue of the SNARC effect as suggested by Reviewer 1. Please see lines 418ff and 641ff. 

We have added on to the discussion, where we – thanks to you – can compare our results with your similar experiment on temporal distance. Indeed, it is very interesting that we both could not find an effect. Please see lines 676ff for the discussion. 

“Large and critical areas of literature are unreferenced. In particular, there is a large literature which has focussed on Chen’s (2013) hypothesis that the obligation to use a future tense for prediction-based FTR should cause speakers to construe future events as more temporally distal, and that this will therefore lead to cross-linguistic effects of FTR grammaticiation on intertemporal decision making. Numerous studies, including ones whose results bear direct relevance to the present MS, have followed up on Chen’s (2013) hypothesis. Yet none of this literature is cited, even though the MS purports to test an idea which can be directly traced to Chen (2013). Reading the MS in its own right, it is difficult to make out why future tense marking should lead to distal temporal construals, which makes it all the more strange that Chen (2013) is left uncited. Chen (2013) provides a closely reasoned and mathematically presented account of this hypothesis...”

Thank you for this very important comment. In hindsight, we should have mentioned Chen’s (2013) research and the LSH hypothesis. The intention was certainly not to leave Chen unreferenced, rather we wanted to experimentally follow-up on our own theoretical recommendations delineated in the article Jäggi et al. (2020) – where Chen’s research was an important part – and to emphasize the direction of our psycholinguistic/cognitive-psychological approach. We understand now, that without having our other article in mind, it seems odd not to reference Chen (2013). You can find the added paragraph on lines 131ff. 

Jäggi, T., Sato, S., Gillioz, C., & Gygax, P. M. (2020). An Interdisciplinary Approach to Understanding the Psychological Impact of Different Grammaticalizations of the Future. Journal of Cognition, 3(1), 10. DOI: http://doi.org/10.5334/joc.100

“Additionally, while the results are interesting, not enough is made of them. For instance, linguists continue to debate the semantics of FTR, and the future tense in particular, and many of these debates revolve around the entanglement modal with temporal notions in future tense semantics. Given the results in the MS indicate there is no effect of tense framing on ratings of temporal distance, this literature should be engaged, which it currently is not. In fact, the MS’s treatment of modality is generally underdeveloped, and this should be addressed.”

Thank you for addressing this point. As already mentioned, we have extended the discussion to include the debated points. Please see lines 676ff. 

“The individual studies are not well motivated. The rationale for each study often does not bear up to close scrutiny, or relies on reference to materials which were not included in the material to be reviewed, i.e. the linguistic task which was developed. This makes judging the substantive contribution and motivation of each study difficult.

This brings up a final point, which is that the methods are not clearly reported. While I know roughly what was done, the precise nature of the empirical work remains a mystery. The methods in their present form would not, for instance, provide enough information for the experiments to be replicated. This obviously needs to be corrected.”

Thank you for this comment. We have added the complete list of items for each experiment on the OSF.io (see https://osf.io/s2axr/?view_only=68a8142718814603b36e3bcb14ff349b). Please see lines 328f.

“I fear this review has been overwhelmingly negative, and I want the authors to know that I do believe they have done some worthwhile work. In this review I attach a copy of their MS which I have annotated with suggestions for how it might be improved, and I hope they will implement these in the future, regardless of the editorial decision from PLOS ONE.

Personally, I am very interested in this work; I am near to completing a PhD which has focussed largely on a very similar research question. Much of my work is presently under review or in prep, but I would be happy to be contacted by the authors should they wish to discuss or share results. I can be

contacted at cole.robertson@ru.nl”

It was very nice of you to have reached out to us and to have let us know about your thesis and the similarity of the subjects. In doing so, we were able to integrate some important results to compare our study in the discussion. Please see lines 676ff.

Response to Reviewer 3:

“Summary

This manuscript presents four experiments exploring the role of different types of linguistic futures on perceptual qualities. Experiment 1 tested French speakers on the placement of event descriptions on a provided timeline. The results revealed no differences in the different types of futures explored, contrary to the hypothesis. Experiment 2 was a replication of Experiment 1, except concrete temporal adverbials were not used, and a more categorical response modality was used. The results were like those of Experiment 1. Experiment 3 was another replication, how this one involved the use of German speakers. The results were like those of Experiments 1 and 2. Experiment 4 was a replication of Experiment 3, but without the present tense items. The results replicated the first three experiments in not showing an effect of different types of futures. The results revealed that the perceptual aspect of future thinking is largely unaffected by the type of future language structure used.”

“Evaluation

This manuscript addresses an interesting topic. The study was well-conduced, and the resulting data are adequately interpreted. Although the study is largely a set of null effects, they do have important theoretical implications, and should be made available. I only have one concern that is detailed below.”

“Minor point

1. I was not entirely satisfied with the description of “perceptual representations”. I think that a more adequate term for capturing what is being referred to here is “embodied representations.””

Thank you for this concern. We address this issue by adding literature on grounded cognition and not referring to mental representations as ‘perceptual representations’ anymore. Please see lines 205ff (the lines refer to the document with tracked changes). 

Response to Reviewer 4:

“I have some points that may require revision of the manuscript. The two main ones relate to the relationship between time and space, and the relation between this study and some parallel work on future discounting. Neither should prevent publication, but the authors may want to think about revisions to make their claims more clear.”

“1.

Is it necessary to include the spatial translation of time in the hypothesis? Experiment 1 asks participants to express the distance on a slider ranging from left (low) to right (high), so a link between the grammaticalisation and perception of distance could appear even if the participants thought of time as flowing from right to left. The left/right mapping might also predict that the effect would be weaker if the slider was in the other direction. But this isn't tested. So I wonder if the spatial mapping is a necessary step in the hypothesis?”

Thank you for this comment. Indeed, we did not test whether the effect would be weaker by changing the slider direction from right to left. However, we still feel that it is important to include the literature on spatial translation of time as it is relevant to understand what we meant when referring to ‘perceptual representations’. We have changed and added a substantial part of the introduction concerning the spatial translation of time regarding grounded cognition. We hope in rewriting this part, it has become clearer why we would like to keep this part in the manuscript. Please see lines 205ff (the lines refer to the document with tracked changes).

“2.

I found it surprising that the paper did not link to the hypothesis by economist Keith Chen on FTR and perceptions of time:

Chen, M. K. (2013). The effect of language on economic behavior: Evidence from savings rates, health behaviors, and retirement assets. American Economic Review, 103(2), 690-731.

Note that the cross-cultural correlation has been criticised:

Roberts, S. G., Winters, J., & Chen, K. (2015). Future tense and economic decisions: controlling for cultural evolution. PloS one, 10(7), e0132145.

But the original idea provides several specific models for how grammaticalisation in language might affect perception of time. In particular, the idea relies on habitual requirements to make distinctions between the future and the present. That is, the more a specific person makes this distinction in their language, the greater the effect on their thought. (that is, there may be within-language differences).

A summary of this research is presented in this preprint (by Cole Robertson and others):

https://papers.ssrn.com/sol3/papers.cfm?abstract_id=3628501

This also includes a summary of recent experimental work on the link between FTR and time perception. Several of these experiments may be relevant to the current study, and the authors may wish to cite them (especially since they claim that little work has been done).”

Thank you for this remark. In hindsight, we should have mentioned Chen’s (2013) research and the LSH hypothesis. The intention was certainly not to leave Chen unreferenced, rather we wanted to experimentally follow-up on our own theoretical recommendations delineated in the article Jäggi et al. (2020) – where Chen’s research was an important part – and to emphasize the direction of our psycholinguistic/cognitive-psychological approach. We understand now, that without having our other article in mind, it seems odd not to reference Chen (2013). You can find the added paragraph on lines 131ff.

Jäggi, T., Sato, S., Gillioz, C., & Gygax, P. M. (2020). An Interdisciplinary Approach to Understanding the Psychological Impact of Different Grammaticalizations of the Future. Journal of Cognition, 3(1), 10. DOI: http://doi.org/10.5334/joc.100

“The preprint above also makes a point about the interpretation of future tense. While many see tense as encoding temporal distance (*when* an event occurs), it also communicates modal possibility (*whether* an event will occur). This is particularly the case for modals (e.g. in English "will" and "might" differ in probability more than they differ in temporal distance).

Cole Robertson, as part of his PhD thesis and a few papers under review, did some studies on FTR and time perception in English, Dutch and German.

One experiment was similar to the present study: Presenting participants with a phrase and asking them to indicate with a slider how far away in time they felt this was referring to. A similar experiment was also done with a slider indicating certainty.

In Robertson's study, within a language, the future tense frame did not predict the rating of temporal distance. This is in line with the current results.

However, the rating of certainty did differ significantly by grammatical form: modals were rated as less likely to occur than the future or present tenses.

Participants in Robertson's study also rated objective times (one month, two months) as a check that participants were rating things sensibly. This also agrees with the analysis of numbers in adverbials in the current study.

In addition, Dutch speakers rated events as more distant than English speakers. A similar thing could be tested in the current study. That is, is it possible to compare the scale scores in the current study for German and French directly? (i.e. the between-language prediction). It seems like the French participants are placing the slider at higher positions on average than the German participants. Isn't this what your hypothesis would predict? I appreciate the items are not exactly the same, but is it still worth noting? I'll also note that the authors are in a good position to do an experiment with bilinguals in several languages, making the comparison more effective (perhaps for future research).

Robertson continued with some experiments to show that an individual person's usage of future vs. modal strategies for talking about probability (collected in a survey similar to Dahl's survey) could be used to predict their attitude to future events (e.g. in a future discounting task).

I appreciate that these studies are not yet publicly available (though I'm sure the author would share their thesis manuscript if asked), and the point is not to undermine the current study. In fact, both lines of research seem to agree in their results. My intention is just to flag this converging evidence, and to ask the current authors whether there is a way of harnessing their current data to investigate the question about modal possibility, in addition to temporal distance.”

We appreciate this information on Cole Robertson’s research a lot and were able to get in touch with them about their thesis. We added the mentioned results to the discussion as well as the insights on modal probability, which we find very interesting. Please see lines 676ff.

While analyzing our results, we actually made the between-language comparison between Experiment 2 and 3. Unfortunately, we did not find a difference between French- and German speakers. We have now added this analysis to the manuscript. Please see lines 541ff.

Due to our experimental design, we cannot analyze the modal possibility from our current data. As all our items use the same sentence structure (adverb, subject, verb, event) and the only modal verb used in some sentences is the German ‘werden’ (to become) as part of the German future tense. We understand that adverbs can also mark possibility (e.g. possible, certain), however, we chose our adverbs to mark temporal information related to different distances from the present to the future. Therefore, we do not see how we could make a claim on modal possibility based on our current data.

“Minor points:

"These structures vary across languages and have shown to

affect how we mentally represent and perceive different aspects of our environment." - This sentence appears to have no evidence attached. It's not clear whether the citations in the previous sentence cover this. To be safe, there are some seminal works that could be easily cited here.”

Thank you for noticing this, we have changed the corresponding sentence. See lines 36f.

“"The grammaticalization of the future, which represents the grammatical manifestations of how to refer to the future, has been scarcely studied with experimental psycholinguistic methods"”

Again, we changed this sentence to clarify that other experimental research has been done. See lines 37ff.

“"As perceptual representations, we mostly refer to perceptual representations of

distance" - Does this mean distance in time? Or are you talking more generally about any kind of domain?”

Thank you for this comment. We realized that our definition of ‘perceptual representations’ was not clear enough. We have changed a great portion of this section. Please see lines 205ff.

“Lines 37-38. It's my understanding that "future time reference" refers to the act of talking about the future, rather than the linguistic devices used to do that?”

Thank you for this question. Future time reference is indeed referred to utterances related to the future. However, FTR entails different linguistic devices. We have changed the corresponding parts to make this clearer. Please see lines 68ff. 

“"FTR varies in the degree of grammaticalization across languages (and at times within language): a low degree of grammaticalization is characterized by adverbials and modal verbs; and a higher degree of grammaticalization is characterized by grammatical structures embedded in the verb, like a suffix of the future tense (e.g., in French: ‘j’ irai à Paris’ [I will go to Paris])."

There is a lot of academic work on grammaticalisation from linguistics, and this could be cited here.

The French example is not illustrative unless one understands French. Please give the interlinear gloss in addition to the translation.”

Thank you for these remarks. We have changed a substantial part of introduction concerning the linguistic basis to make the manuscript clearer. We have used the interlinear gloss for our examples. Please see lines 68ff. 

“"as some languages have been shown to have very low grammaticalized future verb tense and to only use modal verbs to indicate the future tense (e.g., German:"

I'm not sure this is technically correct. German can also use modal modifiers (möglich) or mental state predicates (erwarten). Maybe you're grouping these under modal verbs?”

We appreciate this comment. So far, we were mainly focusing on grammatical aspects of the future tense rather than on modal possibility. Even though, we think this is a very interesting avenue, it goes beyond the scope of the present paper. We have added a bit to the discussion when comparing with Cole Robertson’s results. Please see lines 676ff. 

“"we expect the effect between low and high degree of FTR in German to be somehow stronger than in French". Why "somehow" - you have an explicit hypothesis?”

Thank you for pointing this out. We have adjusted this part in the manuscript. See line 263.

“Table 2 - the significance is easier to interpret if it's explained that PF condition was used as the intercept condition.”

We adjusted this part in the manuscript as well. See lines 372ff. 

“Mixed effects models: You are modelling a scale with a floor and ceiling. Does the model take this into account? It looks like there might be floor effects for the PP condition? You could do this with e.g. logit function.”

Thank you for this comment. We wanted to add a zero-one-inflated beta model to the analyses (it belongs to the family of logit models), as it is recommended for slider data by Vuorre. Unfortunately, our data did not converge – most likely due to not enough data points – so that we could not add this analysis to the manuscript.

Please see Matti Vuorre’s R Blog: https://mvuorre.github.io/blog/posts/2019-02-18-analyze-analog-scale-ratings-with-zero-one-inflated-beta-models/

“Line 365: "it did not significantly account variance for Value" - some missing words in this sentence? Also, some statistical support for this claim should be added (e.g. difference in variance explained in a model comparison test)”

We have adjusted this part to clarify this sentence and have added statistical support. Please see lines 592ff.

The size of the text in the figures is quite small and may not reproduce well at a smaller scale.

Thank you for this remark, we enhanced the text in the figures to make them more readable. Please see images Fig1, Fig2, Fig3, Fig4 and Fig5.

---

## [Decision Letter · Decision Letter 1]

8 Nov 2021

PONE-D-21-18131R1Is the future near or far depending on verb tense markers used? An experimental investigation into the effects of the grammaticalization of the futurePLOS ONE

Dear Dr. Jaeggi,

Thank you for submitting your manuscript to PLOS ONE. After careful consideration, we feel that it has merit but does not fully meet PLOS ONE’s publication criteria as it currently stands. Therefore, we invite you to submit a revised version of the manuscript that addresses the points raised during the review process.

 Thank you for working conscientiously to address all of the reviewers' comments. From a technical/scientific viewpoint, I am happy to inform you that the paper can be accepted for publication. I have, however, processed the paper as 'minor revision' because of the issue raised by reviewer 1 about language errors. Before submitting the final version, can you please make sure you address this point, by for example having your manuscript thoroughly proof-read by the author team and possibly an external native English speaker as well, so that any language issues can be identified and corrected. Otherwise, many congratulations!

We look forward to receiving your revised manuscript.

Kind regards,

Panos Athanasopoulos, Ph.D

Academic Editor

PLOS ONE

Journal Requirements:

Reviewers' comments:

Reviewer's Responses to Questions

**Comments to the Author**

1. If the authors have adequately addressed your comments raised in a previous round of review and you feel that this manuscript is now acceptable for publication, you may indicate that here to bypass the “Comments to the Author” section, enter your conflict of interest statement in the “Confidential to Editor” section, and submit your "Accept" recommendation.

Reviewer #1: (No Response)

Reviewer #3: All comments have been addressed

Reviewer #4: All comments have been addressed

2. Is the manuscript technically sound, and do the data support the conclusions?

Reviewer #1: Yes

Reviewer #3: Yes

Reviewer #4: Yes

3. Has the statistical analysis been performed appropriately and rigorously? 

Reviewer #1: Yes

Reviewer #3: Yes

Reviewer #4: Yes

4. Have the authors made all data underlying the findings in their manuscript fully available?

Reviewer #1: Yes

Reviewer #3: Yes

Reviewer #4: Yes

5. Is the manuscript presented in an intelligible fashion and written in standard English?

Reviewer #1: No

Reviewer #3: Yes

Reviewer #4: Yes

6. Review Comments to the Author

Reviewer #1: I am grateful to the authors for their thorough revision of the manuscript, which I agree is much improved. I have only a one comment, and that is that a number of language errors have crept in since the revision, which need to be addressed as they are reasonably frequent and occasionally a tad jarring.

Reviewer #3: (No Response)

Reviewer #4: Thank you to the authors for responding to my questions. All my concerns have been addressed. The attempt to use the zero-inflated beta model addresses my question about floor effects. Non-convergence is not necessarily a good criteria for omitting this (and in a MCMC paradigm should be avoidable), but it's unlikely that the issue would affect the results.

7. PLOS authors have the option to publish the peer review history of their article (what does this mean?). If published, this will include your full peer review and any attached files.

Reviewer #1: No

Reviewer #3: **Yes: **Gabriel Radvansky

Reviewer #4: No

---

## [Author Response · Author response to Decision Letter 1]

8 Dec 2021

Response to Reviews: Manuscript ID PONE-D-21-18131R1

Response to Reviewer 1:

“I am grateful to the authors for their thorough revision of the manuscript, which I agree is much improved. I have only a one comment, and that is that a number of language errors have crept in since the revision, which need to be addressed as they are reasonably frequent and occasionally a tad jarring.”

Thank you for this comment. We proof-read the manuscript by a professional editor to remove the language errors.

Response to Reviewer 4:

“Thank you to the authors for responding to my questions. All my concerns have been addressed. The attempt to use the zero-inflated beta model addresses my question about floor effects. Non-convergence is not necessarily a good criteria for omitting this (and in a MCMC paradigm should be avoidable), but it's unlikely that the issue would affect the results.”

Thank you for your comment. We reanalyzed the zero-one-inflated beta models for each experiment. Although, we still did not get a viable result for our experiment with three conditions (i.e., Experiment 1, 2 and 3), for Experiment 4 we got the same result as with our mixed effects model (see Figure 1). As we did not get viable results for all experiments, we decided not to include it in the manuscript.

We followed Matti Vuorre’s R Blog for the zero-one-inflated beta model: https://mvuorre.github.io/blog/posts/2019-02-18-analyze-analog-scale-ratings-with-zero-one-inflated-beta-models/

---

## [Editor Report · Decision Letter 2]

5 Jan 2022

Is the future near or far depending on the verb tense markers used? An experimental investigation into the effects of the grammaticalization of the future

PONE-D-21-18131R2

Dear Dr. Jaeggi,

We’re pleased to inform you that your manuscript has been judged scientifically suitable for publication and will be formally accepted for publication once it meets all outstanding technical requirements.

Kind regards,

Panos Athanasopoulos, Ph.D

Academic Editor

PLOS ONE
---

## [Editor Report · Acceptance letter]

10 Jan 2022

PONE-D-21-18131R2 

Is the future near or far depending on the verb tense markers used? An experimental investigation into the effects of the grammaticalization of the future 

Dear Dr. Jaeggi:

I'm pleased to inform you that your manuscript has been deemed suitable for publication in PLOS ONE. Congratulations! Your manuscript is now with our production department. 

Kind regards, 

on behalf of

Professor Panos Athanasopoulos 

Academic Editor

PLOS ONE